# An Automated Tool for Upgrading Fortran Codes

**Lesley Mak [1] and Pooya Taheri [2,3,\*]**

1   Computing Science & Information Systems Department, Langara College, Vancouver, BC V5Y 2Z6, Canada
2   Mechatronic Systems Engineering Department, Simon Fraser University, Surrey, BC V3T 0A3, Canada
3   School of Energy, British Columbia Institute of Technology, Burnaby, BC V5G 3H2, Canada
\*   Correspondence: ptaheri3@bcit.ca

**Abstract:** With archaic coding techniques, there will be a time when it will be necessary to modernize vulnerable software. However, redeveloping out-of-date code can be a time-consuming task when dealing with a multitude of files. To reduce the amount of reassembly for Fortran-based projects, in this paper, we develop a prototype for automating the manual labor of refactoring individual files. ForDADT (Fortran Dynamic Autonomous Diagnostic Tool) project is a Python program designed to reduce the amount of refactoring necessary when compiling Fortran files. In this paper, we demonstrate how ForDADT is used to automate the process of upgrading Fortran codes, process the files, and automate the cleaning of compilation errors. The developed tool automatically updates thousands of files and builds the software to find and fix the errors using pattern matching and data masking algorithms. These modifications address the concerns of code readability, type safety, portability, and adherence to modern programming practices.

**Keywords:** error analysis; Fortran; Python; refactoring; software testing

## 1. Introduction

Software testing identifies any quality or performance issues within the software. In a project environment, testing is used as a tool to provide feedback on the software's current state and to update the system's requirements. Often, this requires significant resources or time to deliver due to the coordination involving testing.

Project assembly such as compiling and building solutions are all necessary tools in executing the implementation and testing phases of a software development lifecycle. Notably, studies reveal that software validation and testing may cost upwards of 50% of the development resources, which indicates how manual code implementation may throttle software development [1,2]. By extension, since code verification must be performed frequently to ensure correctness, this inadvertently contributes to a gradual increase in overhead cost. Defect amplification, defined as a cascading effect of newly generated errors in each developmental step, may be an unavoidable expense if it is left undetected. Errors may cost upwards of three times the cost when periodic reviews are not part of the design [3]. Indubitably, this sort of software testing model is unsustainable in the current market, and it necessitates a more productive solution.

With respect to how resource-intensive testing may be, one approach to this dilemma is to apply automation to improve the testing environment. In this case, there is a variety of study work that demonstrates how automated regression testing can be optimized to fit this criterion. Recent advances in unit testing utilize fault localization [4], selective fault coverage [5], and regression algorithms [6] as a field of focus in automation. To this extent, there is an increasing trend toward automation where developers practice improving the testing quality of the software. According to a survey studying Canadian software companies [7], many of the correspondents automate about 30% of the testing phase and there is a ceiling for the degree of software automation in the testing environment. So, there is a certain reliance on automation, and manual testing is still frequently used to cover testing exceptions.

### 1.1. Software Testing

There is a current demand to automate diagnostic tools such as a static analyzer [8], which partially builds a framework that can detect semantic errors for the user, and direct users to find the underlying errors in the post-analysis. There are commercially available modular solutions [9,10] which apply applications to check for syntax errors. These frameworks have features such as visual graphic interfaces that highlight the issues and identify potential variables in their rigorous testing. There is a current demand to create an automated system that can determine compilation errors and reduce the time used to refactor those results.

An alternative approach includes applying test case data generation to assess the structural integrity of the data coverage as it iterates through each subroutine junction. As data coverage in testing involves defined specifications to validate with its system requirements, there is a limit to the quantity of software testing depending on the scope of the testing method. A dynamic approach to this dilemma is to implement path selection in which the program processes through selective paths to assess the data coverage [11]. With this method, test generated data are used to traverse through each junction until each path has been attempted. Conversely, if a blocked path is encountered, this approach would recursively navigate until a proper path is found.

### 1.2. Fortran

Fortran was one of the first high-level programming languages intended for mathematical computations used in sciences and engineering. The intuitive abstraction of mathematical procedures enabled rapid development of numerical solutions to scientific problems, at a time when most programs were still hand-coded in assembly language. FORTRAN 77 built up a huge legacy, and many coding projects were developed using it and continue to use it to this day. Since its first release, five Fortran standards have been released. Many of the computer models used in scientific research have been developed in Fortran over many years [12–14].

The Fortran language retains its high performance through its array-oriented design, strong static guarantees, and its native support for shared memory and distributed memory parallelism. High-Performance Fortran (HPF) provides high-level parallelization leading to shorter runtime and higher efficiency [15]. There were about 14,800 citations in Google Scholar mentioning FORTRAN 77 between 2011 and 2021, showing that Fortran is not obsolete as many predicted it to be. Fortran is attractive to scientists because of its high-level array support, low runtime overhead, predictable and controllable performance, ease of use, optimization, productivity, portability, stability, and longevity [16–18].

However, since the FORTRAN 77 language was designed with assumptions and requirements very different from today's software requirements, code written using it has inherent issues with readability, scalability, maintainability, type safety, and parallelization. The evolutionary process of code development means that models developed in Fortran often use deprecated language features and idioms that impede software maintenance, understandability, extension, and verification. As a result, many efforts have been aimed at refactoring legacy code to address these issues [16,17].

### 1.3. Fortran Refactoring

Refactoring is the process of changing a software's internal structure while preserving its external behavior [19]. Refactoring is usually performed using various operations, including renaming attributes, moving classes, replacing obsolete structures, splitting packages, and parallelizing the code [20]. Refactoring allows modification of code artefacts to address new system requirements [21]. The general benefits of refactoring can be listed as [22]:

- Improving readability and quality;
- Reducing the maintenance requirements;
- Facilitating design/interface changes;

- Avoiding poor coding practices;
- Removing outdated constructs;
- Increasing performance and speed; and
- Parallelizing the code [23].

However, if not executed properly, refactoring can cause security risks due to undesirable changes [24]. Preserving code's behavior is the biggest challenge of automated refactoring [25]. To successfully refactor a system, we need to do it in small steps, while designing proper tests to make sure that the behavior of the code is not negatively affected by internal changes [22].

Fortran has an emphasis on backward compatibility [26]. Anachronistic features used in older versions worsen code readability and performance [21]. Fortran refactoring makes the code easier to understand and maintain, while optimizing its performance and portability [27].

Therefore, we advocate for the refactoring of legacy code as languages evolve to avoid sedimentary programs with layers of past and present code and upgrading Fortran code to a modern form, eliminating deprecated features and introducing structure data types. This assists the maintenance, verification, extension, and understandability of code. These upgrades produce a more readable Fortran program that includes safeguards to prevent accidental mistyping of variables and unintentional changes in named constants during program execution.

In this paper, we focus on a set of modifications to remove the inconsistencies and troublesome structures that exist in a software evolved through several versions of Fortran. These modifications address the concerns of code readability, portability, and adherence to modern programming practices. FORTRAN 77 programs can be made entirely type safe through program transformations resulting in fewer errors. There are a number of restructuring and refactoring tools for Fortran [12–17,21–23,25–36]. Some of the modifications provided in the literature for a modern Fortran code are listed in Table 1 citing the corresponding references.

**Table 1.** List of refactoring operations to upgrade a Fortran code.

| Refactoring Modification | References |
|---|---|
| Removal of IMPLICIT typing | [17,22,33,35,36] |
| COMMON block elimination | [16,17,22,33] |
| EQUIVALENCE statement elimination | [16,17] |
| Change fixed form to free form | [33,35,36] |
| Conversion of fixed-value variables to parameters | [33,35,36] |
| Removal of GLOBAL variables | [33,35] |
| Use of the INTENT attribute | [17,33,35] |
| Replacement of REAL variables with CHARACTER variables | [33] |
| Elimination of arithmetic IF statements | [22,33,36] |
| Replace old-style DO loops and use END DO statements | [21,22,27,33,36] |
| Use of allocatable arrays, CASE construct, and structures | [33,35] |
| Clear commenting and time/date stamping | [33,35] |
| Elimination of GO TO statements | [22,33,36] |
| Use of individual subroutine files | [21] |
| Remove tabs | [35] |
| Replace obsolete operators and unreferenced labels | [36] |
| Replace statement functions with internal functions | [33] |

### 1.4. Project Description

The ForDADT (Fortran Dynamic Autonomous Diagnostic Tool) project is designed to reduce the amount of refactoring necessary when compiling Fortran files. In the case where there are programs that access many files, compilation time becomes exponentially costly for developers. Rather than focusing on automating the entire environment, the ForDADT tool is developed to partially automate common error cases from the Fortran program. This project has been developed to bridge the refactoring and automation processes to help improve developers' efficiency when refactoring code. ForDADT is open source, and the codes are available at https://github.com/LMAK00/fordadt.

Figure 1 demonstrates the manual algorithm for updating outdated syntax in the system structure. This necessitates a considerable number of resources to meticulously apply the appropriate fix to each component. It may be explicitly difficult to approach a solution if these syntax errors scale in proportion to the size of the program file.

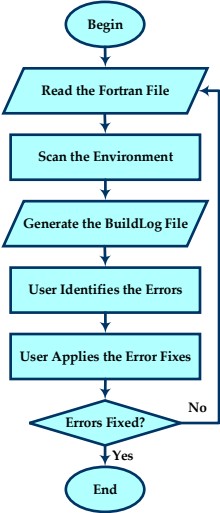

**Figure 1.** Flow chart of manual testing for error cases.

The goal of this design is to develop an automated process of upgrading software codes composed of different Fortran version standards and adapt those legacy source codes to a modern design. The ForDADT project removes Fortran errors and updates the codes written in older versions to version 19.0.5.281. The developed tool automatically updates thousands of Fortran files and builds the software to find and fix the errors using pattern matching and data masking algorithms.

A current dilemma with most available compilers is that they can detect several errors before responding with a "catastrophic error" and terminating the system build. In the case where a file has million lines of code, detecting a few lines of syntax errors at a time is not a suitable solution if the user must recompile the application a multitude of times until the file meets the build requirements. We tested our tool using a real commercial software including thousands of files and millions of lines of code and evolved throughout decades. The manual and file-based algorithms are impractical and prone to errors for software of this scale.

The project is written in Python which checks for certain pattern matches in the Fortran files to report the errors found. Every software update will be made in a series of steps [12]:

1.  Identify and save the current program;
2.  Apply a specific update;
3.  Verify the new program version by comparison with the previous one;
4.  Accept/reject the change; and
5.  Document the change.

In its current form, it can detect some of the common errors and produce a general set of solutions to solve those issues. The use of Python with Fortran and interfacing them together has been realized successfully using different tools such as f90wrap and F2PY [37,38]. ForDADT operates based on two subcomponents:

1. A file reader that identifies errors and creates an error log file; and
2. A file solution which extracts the error log file and fixes the issues of files.

The purpose of this paper is to develop an automation tool that can identify the relative syntax errors in the file and implement a suitable patch for each element and eliminate the necessity to continuously recompile the solution.

## 2. Methodology

A common issue when porting legacy Fortran source files to a modern compiler is that the previous coding standard may not completely conform with the current practices. There are cases where implementing one of the following changes listed in Section 1.3 may cause a cascading effect of compiler errors. For example, when a file has million lines of code, manually correcting a few lines of syntax errors at a time is not a suitable solution if the user must recompile the application a multitude of times until the file satisfies the build requirements. To this extent, there is an opportunity to apply analytic tools to improve the quality control testing and minimize the manual code tracing in the system build.

Findings in related works reveal that automation has a net positive effect in reducing the project's timeframe and cost [2]. There is a significant improvement in quality control resulting from reducing the number of errors in the testing process. This approach requires a manual process of reworking solutions to fix the issues. The ForDADT project was developed to challenge the possibility of automation in this sort of model.

This project can be divided into two fundamental parts: (1) error checker and (2) code analysis. ForDADT addresses obsolete Fortran statements by extracting the necessary source code segments as partial input data for the code analysis to process. The focus on this design is to mitigate the degree of programmer intervention with the Python program. Figure 2 shows the project flow diagram.

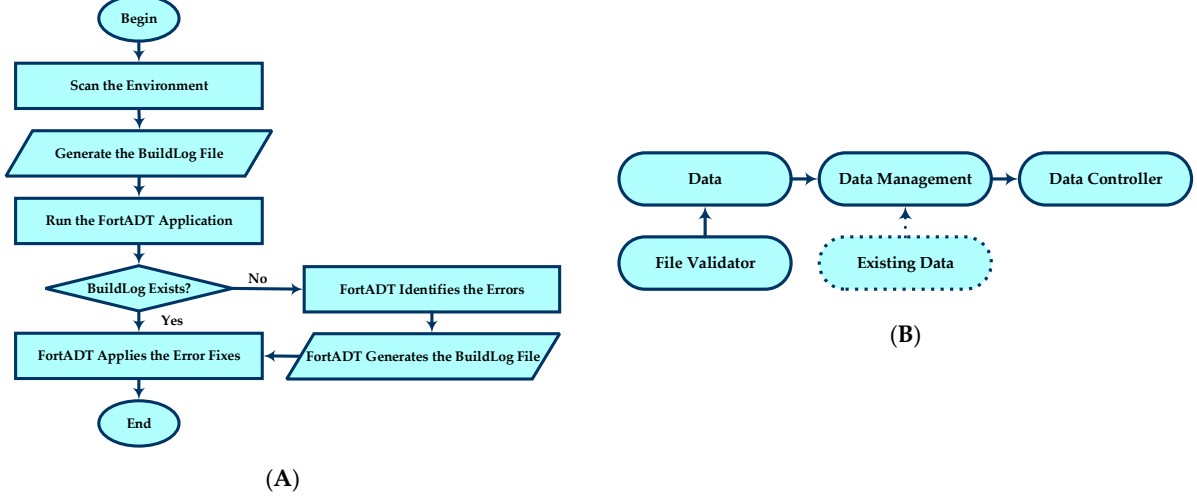

**Figure 2.** (**A**) Flow chart of the project flow. (**B**) Overview of the project diagram for ForDADT.

With automation, ForDADT strives to improve productivity for program developers by automating the debug process cycle to reduce the amount of necessary refactoring. Menial errors can be corrected to a standardized template which helps with code readability as well as ensuring that the errors encountered in the previous iteration are properly fixed. Table 2 compares the features of manual and automated testing operations for basic and complex tasks.

**Table 2.** Cross-analysis between manual and automated testing operation.

| Task | Manual Operation | Automated Operation |
|---|---|---|
| Basic | Simple fix to clean the error<br>Easy to trace the error file | Standardized solutions<br>Fixes all errors at once<br>Optimum for large file compilations |
| Complex | Time consuming to check<br>May cause cascading issues<br>Difficult to trace for large files | Time consuming to scan every file<br>Automated corrections are explicitly stated<br>May not be reliable for non-standard operations |

### 2.1. Build Path

The project relies on an output file produced by the Intel® Fortran compiler which natively identifies the build error in the output terminal. The BuildLog file produced by the compiler is then used as a reference to identify the errors in the current build. The initial process for error checking is to extract the relevant data from the output file. If a BuildLog file is provided by the Integrated Development Environment (IDE), the program will run the html2text [11] application to convert the data into a readable text file for the automation process. The html2text application extracts meaningful error data from the IDE's source html output file and translates the html file to a text file for post-processing. If a BuildLog file does not exist, a corresponding BuildLog compatible with the program will be provided.

### 2.2. Error File Automation

This initial step generates a pseudo-BuildLog file which stores the resulting data for later processing. Pattern matching algorithms are used to identify Fortran coding patterns and determine the existing errors in the files. With post-pattern matching analysis, this framework reduces the necessity of a native compiler and acts as a standalone program.

The program runs calculations to compile the presented data into proper Fortran software syntax. A subroutine is run to match the error cases from the analysis which searches for the individual file and the file index. There are variations in the error case categories, but all calculations rely on detecting the provided error case and applying an appropriate solution for each individual error.

The pattern-matching step uses regular expressions to note the intrinsic functions denoted in the Fortran compiler. In the current state, the error file can detect common syntax errors that are not caught when porting over to Intel®Visual Fortran Compiler (ifort) and flag the syntax errors for post-processing in the automation process. The generated BuildLog file is a short form version of the existing template produced from the Visual Studio IDE. For continuity purposes, the internal and standalone BuildLog files are both expressed in a similar template so that the standalone application can digest input files into the automation process.

The resulting code from the following extraction is a set of variables to be checked for validity. An example of this is the variable validation step in the post-analysis of the automation process. The previous implementation of the Fortran compiler did not re-quire variable declaration. In contrast, this is a required element for the builder in the current implementation.

Table 3 shows the description of the encountered compiler errors. For each syntax error, ForDADT analyzes the properties of the source code if there are conflicting elements. Examples are provided to show the common issues that may occur in the code. The error/warning causes are highlighted in red.

**Table 3.** List of common syntax errors from Fortran compiler with the causes highlighted in red.

| Code | Compiler Error/Warning Message | Example |
|---|---|---|
| 6186 | This character is not valid in a format list. | FORMAT(A5,TL,J) |
| 6222 | This IMPLICIT statement is not positioned correctly within the scoping unit. | SUBROUTINE<br>INTEGER XL<br>IMPLICIT NONE |
| 6239 | This is an invalid DATA statement object. | SUBROUTINE<br>DATA XL |
| 6278 | This USE statement is not positioned correctly within the scoping unit. | SUBROUTINE<br>INTEGER XL<br>USE LIBRARY |
| 6362 | The data types of the argument(s) are invalid. | CALL(XR) |
| 6401 | The attributes of this name conflict with those made accessible by a USE statement. | USE MFI, ONLY: XL<br>INTEGER XL |
| 6404 | This name does not have a type, and must have an explicit type. | SUBROUTINE<br>IMPLICIT NONE<br>XL = 1 |
| 6418 | This name has already been assigned a data type. | INTEGER XL<br>. . .<br>INTEGER XL |
| 7319 | This argument's data type is incompatible with this intrinsic procedure; procedure assumed EXTERNAL. | CALL(XR) |

## 3. Implementation

This section details ForDADT's implementation of the automation procedure. The error analysis executes by extracting individual lines of code from the existing Fortran files and scans each line of code for certain keys used in its verification assessment. The initial search extracts these Boolean flags and stores them in a routine checklist that notes the priority order of the lines of code. Each line of code must consider the argument flags in the checklist such as continuation lines, Fortran spacing rules, and a variety of general Fortran code structures for ForDADT to diagnose the compiler errors in the automation step. For instance, to identify Fortran declaration flags, the program uses Python's built-in text processing modules to denote the string literals of each line of code. The program uses string-searching algorithms such as regular expressions (regex) [39] to search for string patterns which are then separated into meaningful segments for diagnosis. Using this approach, the program can detect key violation occurrences in the routine check and send the appropriate data to the error file. To illustrate the pattern matching procedure, a simplified example of this design is visualized with the following Figure 3.

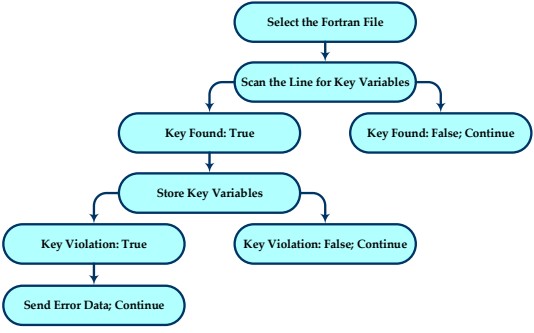

**Figure 3.** Simplified tree diagram of the verification assessment phase.

To understand how the algorithms are applied to identify the syntax errors in Fortran's coding language, the Fortran code analysis can be sorted into separate compositions: compiler directive alignment, variable verification, and syntax correction. In this step, ForDADT scans for local Fortran type files for code analysis. If an appropriate Fortran file extension (.f, .f90, and .for) is found, the analyzer checks for indexing flags that are incompatible with the current Fortran system requirements.

### 3.1. Compiler Directive Alignment

A prior feature of Fortran type casting defaults variable statements into INTEGER and REAL arguments depending on the initial letter of the variable. This previous design is inconvenient since it may lead to unexpected compiler behavior from mistyped declarations. The "IMPLICIT NONE" statement declaration can remedy this issue by explicitly defining variables in the source code, preventing variable conflicts from occurring. Consequently, all implicitly declared data types must be specified in the local file. As a result, the syntax order must be reordered to accommodate these changes. An example of this is given in Figure 4, which presents how the algorithm manages improper indexing in the local source code. The order of the precedent indices is crucial to the Fortran subprogram syntax such that the statements must be ordered in the following sequence: subroutine block, "IMPLICIT NONE" statement, and module "USE" statement. The algorithm logs each index of the above statements and conditionally flags those misplaced indices for post-analysis.

| **Algorithm 1: Finding Fortran error 6278** |
|---|
| 1     *lines* ← list of Fortran file's content |
| 2     **for** *i* in *lines* **do** |
| 3       /* The "*" is used to catch comments */ |
| 4       **if** *lines*[*i*] contains "use" and not "*"' **then** |
| 5         *useIndex* ← *i* |
| 6       **else if** *lines*[*i*] contains "IMPLICIT NONE" **then** |
| 7         *implicitIndex* ← *i* |
| 8       **end if** |
| 9       **if** *lines*[*i*] contains "SUBROUTINE" **then** |
| 10        *subroutineIndex* ← *i* |
| 11        **if** *useIndex* < *implicitIndex* and *implicitIndex* ≠ *subroutineIndex* **then** |
| 12          write "error #6278" to file |
| 13        **end if** |
| 14        *useIndex* ← *i* |
| 15        *implicitIndex* ← *i* |
| 16       **end if** |
| 17     **end for** |

**Figure 4.** The algorithm for finding Fortran compiler error 6278 (USE statement positioned incorrectly).

### 3.2. Variable Verification

The implementation of the variable verification algorithm is a critical step for ForDADT. To properly differentiate procedure statements in the source file, variable identification is necessary to classify Fortran declaration statements into valid variables defined by the local file. In this scenario, all variables must be explicitly declared due to the "IMPLICIT NONE" declaration, whereas any undeclared variables will be flagged in the error output file. Figure 5 shows the pseudocode of the extraction process used to detect variable declarations in the following steps.

| Algorithm 2: Finding Fortran error 6404 |
|---|
| 1   *varList* ← to store declared variables |
| 2   *exList* ← to store library variables |
| 3   **for** *i* in range of Fortran file **do** |
| 4       *A* ← *list*[*i*] |
| 5       **if** "C" or "*" or "!" in *A* **then** |
| 6           skip the comment line |
| 7       **end if** |
| 8       **if** variable declaration in *A* **then** |
| 9           *varList* ← *internal variable declarations* |
| 10          *exList* ← *external variable declarations* |
| 11      **end if** |
| 12      strip all the excess data from files |
| 13      *varIndex* ← 0 |
| 14      **while** *varIndex* ≠ end of line **do** |
| 15          **if** *varIndex* ≠ *varList* and *varIndex* ≠ *exList* **then** |
| 16              write "error#6404" + *varIndex* to file |
| 17          **end if** |
| 18          increment *varIndex* |
| 19      **end while** |
| 20  **end for** |

**Figure 5.** The short-form algorithm for finding Fortran compiler error 6404 (explicit type missing).

### 3.2.1. Variable Generation

The initial search scans for local variable statements and stores the named variable into a list. This is accomplished by using regex string manipulation to identify potential variables in the file, in which each line of code is flagged when a declaration (e.g., INTEGER, REAL, . . . ) is found. External linking files such as module subprograms also need to be traversed to capture any variable statements declared in the subprogram. If an external file declaration statement is found in the local file, ForDADT's readError subroutine executes the search for the selected external file in the root directory and produces a BuildLog file identical to the error file. The code then transverses through the external file and adds the external type declarations to the variable generation list from the local Fortran file using addLibraryVariables and addLibraryFiles functions. Figure 6 shows a snapshot of a BuildLog file showing extrinsic and undeclared variables and their location.

```
1   BuildLog·for·Fortran·Converter·
2   Compiling·with·Intel(R)·Visual·Fortran·Compiler·19.0.5.281·
3        ·-·backup\SIME1.for(161)·error·#6404:·[MaxCls]
4        ·-·backup\SIME1.for(166)·error·#6404:·[Nmach]·
5        ·-·backup\SIME1.for(181)·error·#6404:·[Isle]·
6        ·-·backup\SIME1.for(229)·error·#6404:·[Nmach]·
7        ·-·backup\SIME1.for(264)·error·#6404:·[Kpar]·
8        ·-·backup\SIME1.for(361)·error·#6404:·[M]·
9        ·-·backup\SIME1.for(367)·error·#6404:·[Locked]·
10       ·-·backup\SIME1.for(370)·error·#6404:·[LifeTime]·
```

**Figure 6.** Snapshot of a BuildLog file generated by checkError6404 that shows the location of extrinsic and undeclared variables.

Figure 7 shows an example of removal of IMPLICIT NONE and explicit declaration performed by ForDADT in one of the software files.

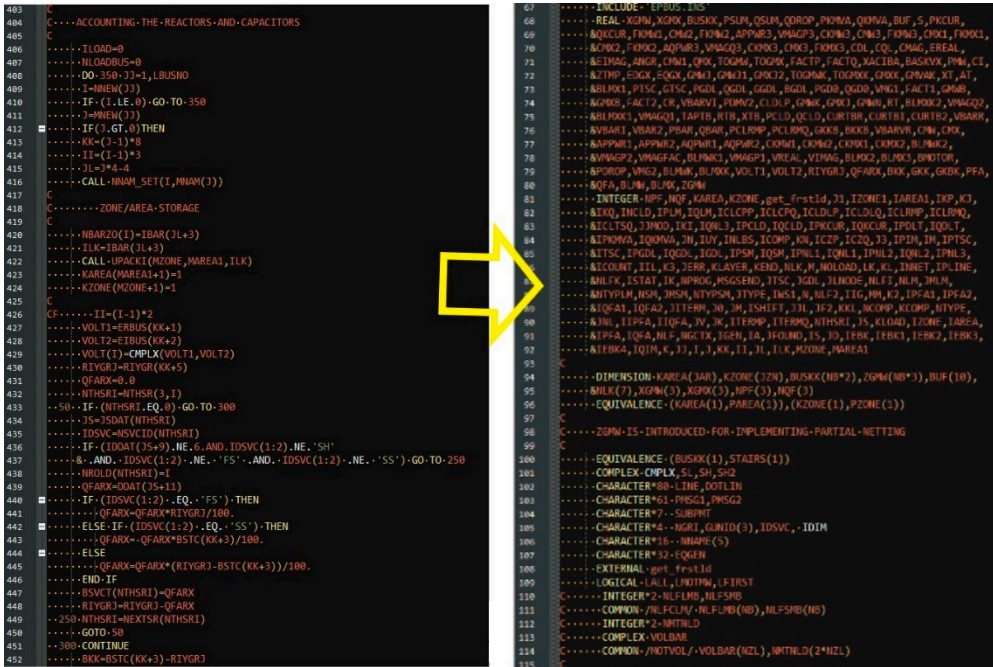

**Figure 7.** Snapshot of a sample program showing the process of removing implicit declaration and adding external variable declarations.

3.2.2. Selective Data Masking

The next phase is to remove the implicit function calls from the source file. These implicit functions must be specifically pruned due to exception function parameters that are not caught by the authentication process. An example of this is parameter settings such as format descriptors which are implicitly understood by the Fortran compiler but not by ForDADT. ForDADT performs selective function masking to remove these function calls from the verification process. As the function calls are not required in this set of algorithms, these functions must be mitigated from the computation data to prevent malformed variable declarations in the detection phase.

The variable verification addresses this problem by procedurally stepping each line of code to prune the function calls from the automation process. For instance, Figure 8 shows a representation of how the function masking works for the verification step. The procedure iterates in reversive order and tracks the string indices of the logic and mathematical operators. If a closed parenthesis is followed by a function name, then the function call will be removed from the line of code. The operator characters are used as a condition statement to detect the beginning of the function name to assist in this process.

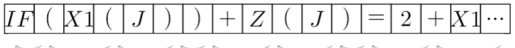

**Figure 8.** Procedural stepping algorithm for selective data masking.

A sample program in Figure 9 demonstrates the pruning process that occurs during the selective data masking. Data masking hides specific data elements inside dataset [40]. With the selective data masking mentioned above, the process considers Fortran internal attributes such as user comments, conditional statements, and compiler directive statements in the regex's case matching algorithm. This masking procedure is illustrated in three steps:

1.  The initial regex pattern matching [41] that prunes the internal directives;
2.  The program truncates the string objects into legal identifiers; and
3.  The verification phase that matches the variable lines (e.g., N, M, FIB1, FIB2, FIBN in Figure 9) to the lines of code.

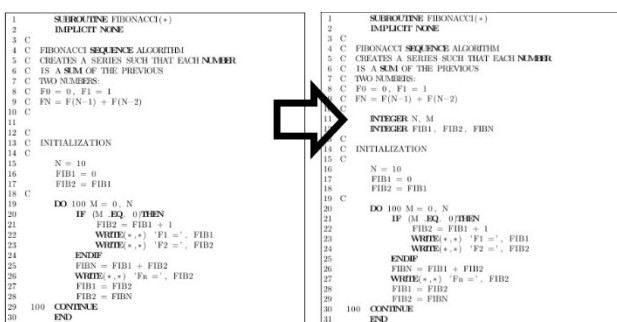

**Figure 9.** Fibonacci subroutine written in FORTRAN 77 and the pruned results.

### 3.2.3. Variable Verification

In the variable verification phase, the stored variables are used to compare each string in the file. The match list includes the variable generation from the above phase as well as a list of intrinsic Fortran variables and functions to be cross-matched in the verification process. An intrinsic list is necessary in the verification step because ForDADT must oversee references to the functions undetected by the data masking algorithm. After cycling through the exhaustive list of keywords such as internal functions, declaration identifiers, and user-defined keywords, any undeclared variable detected is sent to the error file for processing. Figure 10 shows a list of intrinsic functions and keywords stored as a tuple.

```
##          DOESNT HAVE ALL THE OBSOLETE FUNCTIONS SUCH AS INUM, JNUM, KNUM, ...
reservedNotepadKey = ['__FILE__', '__LINE__', '__DATE__', '__TIME__', '__TIMESTAMP__', 'ABS', 'ACCESS', 'ACHAR', 'ACOS', 'ACOSD', 'ACTION', 'ADJUSTL', 'ADJ
'ATAN2D', 'ATAND', 'BACKSPACE', 'BIND', 'BIT_SIZE', 'BITEST', 'BITL', 'BITLR', 'BITRL', 'BJTEST', 'BKTEST', 'BLANK', 'BLOCKDATA', 'BREAK', 'BTEST', 'CABS',
'CSHIFT', 'CSIN', 'CSQRT', 'CYCLE', 'DABS', 'DACOS', 'DACOSD', 'DASIN', 'DASIND', 'DATA', 'DATAN', 'DATAN2', 'DATAN2D', 'DATAND', 'DATE', 'DATE_AND_TIME',
'DMIN1', 'DMOD', 'DNINT', 'DO', 'DOT_PRODUCT', 'DOUBLE', 'DOUBLECOMPLEX', 'DOUBLEPRECISION', 'DOWHILE', 'DPROD', 'DREAL', 'DREAL', 'DSIGN', 'DSIN', 'DSIND'
'ENDTYPE', 'ENDWHERE', 'ENTRY', 'ENUM', 'EOR', 'EOSHIFT', 'EPSILON', 'EQUIVALENCE', 'ERR', 'ERRMSG', 'ERRSNS', 'EXIST', 'EXIT', 'EXP', 'EXPONENT', 'EXTERNAL
'ID', 'IDATE', 'IDENTIFIER', 'IDIM', 'IDINT', 'IDNINT', 'IEOR', 'IF', 'IFIX', 'IIABS', 'IIAND', 'IIBCLR', 'IIBITS', 'IIBSET', 'IIDIM', 'IIDINT', 'IIDNNT',
'INTRUP', 'INVALOP', 'IOLENGTH', 'IOMSG', 'IOR', 'IOSTAT', 'IOSTAT_MSG', 'IQINT', 'IQNINT', 'ISHA', 'ISHC', 'ISHFT', 'ISHFTC', 'ISHL', 'ISIGN', 'ISNAN', 'I
'KIND', 'KINT', 'KIOR', 'KISHFT', 'KISHFTC', 'KISIGN', 'KMAX0', 'KMAX1', 'KMIN0', 'KMIN1', 'KMOD', 'KNINT', 'KNOT', 'KZEXT', 'LACFAR', 'LBOUND', 'LEADZ', '
'NAMELIST', 'NARGS', 'NBREAK', 'NDPERR', 'NDPEXC', 'NEAREST', 'NEXTREC', 'NINT', 'NML', 'NONE', 'NOT', 'NULLIFY', 'NUMBER', 'NUMBER_OF_PROCESSORS', 'NWORKEI
'QABS', 'QACOS', 'QACOSD', 'QASIN', 'QASIND', 'QATAN', 'QATAN2', 'QATAND', 'QCMPLX', 'QCONJG', 'QCOS', 'QCOSD', 'QCOSH', 'QDIM', 'QEXP', 'QEXT', 'QEXTD', '
'RETURN1', 'REWIND', 'REWRITE', 'RRSPACING', 'RSHIFT', 'SAVE', 'SCALE', 'SCAN', 'SECNDS', 'SEGMENT', 'SELECT', 'SELECTCASE', 'SELECTED_INT_KIND', 'SELECTED_
'THEN', 'TIMER', 'TINY', 'TO', 'TRANSFER', 'TRANSPOSE', 'TRIM', 'TYPE', 'UBOUND', 'UNDFL', 'UNFORMATTED', 'UNION', 'UNIT', 'UNLOCK', 'UNPACK', 'USE', 'VAL'
##compiler directives, not used
reservedCD = ['ALIAS', 'ASSUME_ALIGNED', 'ATTRIBUTES', 'DECLARE', 'DEFINE', 'DISTRIBUTE POINT', 'ELSE', 'ELSEIF', 'ENDIF', \
   'FIXEDFORMLINESIZE', 'FREEFORM', 'IDENT', 'IF', 'IF DEFINED', 'INTEGER', 'IVDEP', 'LOOP COUNT', 'MEMREF_CONTROL', 'MESSAGE', 'NODECLARE' \
   'NOFREEFORM', 'NOPARALLEL', 'NOOPTIMIZE', 'NOPREFETCH', 'NOSTRICT', 'NOSWP', 'NOUNROLL', 'NOVECTOR', 'OBJCOMMENT', 'OPTIMIZE', 'OPTIONS', \
   'PACK', 'PARALLEL', 'PREFETCH', 'PSECT', 'REAL', 'STRICT', 'SWP', 'UNDEFINE', 'UNROLL', 'VECTOR ALIGNED', 'VECTOR ALWAYS', 'VECTOR NONTEMPORAL', \
   'VECTOR UNALIGNED']
reservedOPSpecifiers = ['.FALSE.', '.TRUE.', 'ACCESS', 'ACTION', 'APOSTROPHE', 'APPEND', 'ASIS', 'ASSOCIATEVARIABLE', 'ASYNCHRONOUS', \
   'BIG_ENDIAN', 'BINARY', 'BLOCKSIZE', 'BUFFERCOUNT', 'BUFFERED', 'CARRIAGECONTROL', 'COMMA', 'COMPATIBLE', 'CONVERT', 'CRAY', 'DECIMAL', \
   'DEFAULT', 'DEFAULTFILE', 'DELETE', 'DELIM', 'DENTNONE', 'DENYRD', 'DENYRW', 'DENYWR', 'DIRECT', 'DOWN', 'ENCODING', 'ERR', 'FDX', 'FGX', \
   'FILE', 'FIXED', 'FORM', 'FORMATTED', 'FORTRAN', 'IBM', 'IOFOCUS', 'IOSTAT', 'ISTAT', 'KEEP', 'LIST', 'LITTLE_ENDIAN', 'MAXREC', 'NAME', \
   'NATIVE', 'NEAREST', 'NEW', 'NEWUNIT', 'NO', 'NONE', 'NOSHARED', 'NULL', 'OLD', 'ORGANIZATION', 'PAD', 'PLUS', 'POINT', 'POSITION', 'PRINT', \
   'PRINT/DELETE', 'PROCESSOR_DEFINED', 'QUOTE', 'READ', 'READONLY', 'READWRITE', 'RECL', 'RECORDSIZE', 'RECORDTYPE', 'RELATIVE', 'REPLACE', \
   'REWIND', 'ROUND', 'SCRATCH', 'SEGMENTED', 'SEQUENTIAL', 'SHARE', 'SHARED', 'SIGN', 'STATUS', 'STREAM', 'STREAM_CR', 'STREAM_LF', 'SUBMIT', \
   'SUBMIT/DELETE', 'SUPPRESS', 'TITLE', 'TYPE', 'UNFORMATTED', 'UNIT', 'UNKNOWN', 'UP', 'USEROPEN', 'UTF-8', 'VARIABLE', 'VAXD', 'VAXG', \
   'WRITE', 'YES', 'ZERO']
```

**Figure 10.** List of intrinsic functions and keywords used for variable verification.

### 3.3. Syntax Correction

The syntax correction phase of the research work relies on the created error file. The error file indicates the source file, index line, and the variable declaration statement where the issue occurs. Once ForDADT acknowledges the corresponding data from the error BuildLog file, it checks for match cases for the appropriate solution in the application. For instance, to detect Fortran error #6404, ForDADT scans through the error file until it detects a text line with the code test.for(35) error #6404: [var1], and extracts the relevant data. ForDADT executes an appropriate solution based on the Fortran file (test.for), line number (35), error code (6404), and variable (var1). This retrieved result is then matched with the appropriate error code and inserts the correct declaration identifier into the variable block that removes the obstructing issue detected by ForDADT.

## 4. Algorithms

The ForDADT validation method searches each Fortran file with a set of execution rules specific to the Fortran compiler error in question. The typical case structure in this error detection phase involves two different scenarios: (a) general compiler directives and (b) specific syntax errors for which the project uses string matching to analyze the target file. The code extraction process generates a set of requirements appropriate to the scope of the target compiler error. For instance, the error detection considers two case types (directives or position rules) of compiler errors. This includes misplaced directive declarations that do not follow the order precedence for variable declarations which are flagged for error analysis in the code extraction process.

The main code that runs the program checks if a BuildLog exists in the file directory and runs the appropriate subroutines. The html2text [11] is necessary to convert the generated BuildLog file from the Visual Studio IDE.

Figure 11 illustrates a sample of the directive compiler flags that are passed to For-DADT. The proper order of precedence is as follows: (1) function declaration block (subroutine), (2) implicit typing (IMPLICIT NONE), and (3) external module importing. The automation process can note erroneous declarations and allocate resources to fix the issues using Boolean flags. Misplaced declarations are flagged by ForDADT, which updates the BuildLog file used for postprocess identification.

```
1           SUBROUTINE FIBONACCI(*)
2           IMPLICIT NONE
3           USE FORMULA1
4           INCLUDE 'ARGS1.INS'
5           INCLUDE 'ARGS2.INS'
            …
21          INTEGER N, M, FIB1, FIB2, FIBN
            …
51    100   CONTINUE
52          END
            …
61          IMPLICIT NONE
62          SUBROUTINE SUB_A(*)
63          INCLUDE 'ARGS3.INS'
64          USE FORMULA2, ONLY: FCALC
            …
71          IF(FLAG .EQ. 0) THEN
72             CALL CALC(NI)
73          ENDIF
74          FORMAT (5X,'String', F8.6)
75          END
```

**Figure 11.** Sample program with Boolean flags marking proper placement. The blue font indicates flagged compiler directives and red font marks incorrect placement.

The Readerror.py subprogram does not rely on the IDE to produce an error file but recreates an identical BuildLog file (Figure 12). To facilitate a substitute error file, it is necessary to use pattern matching analysis to process the error output file from the program.

With the converted BuildLog text file, the subprogram calls variations of fixError#### subroutines to fix the files. Each subroutine initializes by checking for a match case in the text file; for instance, it checks to find "error #6278", which notifies the program of which Fortran file and index line where the issue occurs. In the case of subroutine, fixError6278, it checks for a misplaced "IMPLICIT NONE" call in the Fortran file and realigns the statement to the correct index line. Figure 13 demonstrates the pseudocodes for cleaning compiler errors 6278, 6418, 6362-7319, and 6222. The required inputs to the codes are the directory and list of the Fortran files. Precondition of the codes are existence of the Fortran files in the directory and proper format of BuildLog file.

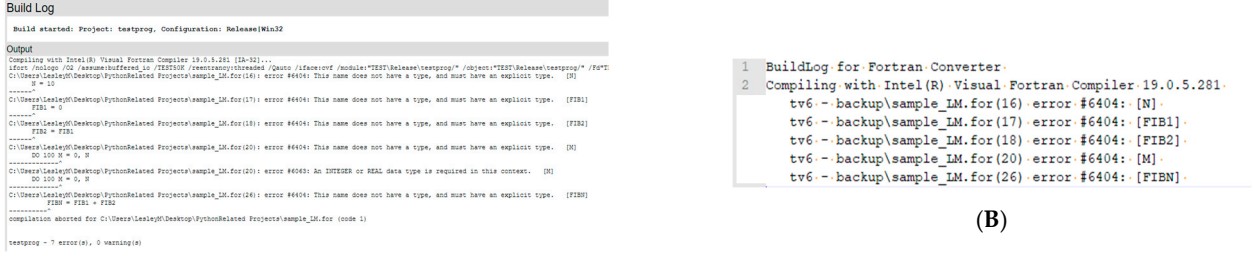

**Figure 12.** (**A**) Typical error output BuildLog html file. (**B**) BuildLog text output from ForDADT.

| (A) Algorithm 3: Fixing Fortran error 6278 | (B) Algorithm 4: Fixing Fortran error 6418 |
|---|---|
| 1   **for** *i* in range of Fortran file **do** | 1   **for** *i* in range of Fortran file **do** |
| 2     *A ← list[i]* | 2     *A ← list[i]* |
| 3     *Index ← A*.find("error#6278") | 3     *Index ← A*.find("error#6418") |
| 4     **if** *index* ≠ −1 **then** | 4     **if** *index* ≠ −1 **then** |
| 5       *filename* ← Fortran file name from error line | 5       *filename* ← Fortran file name from error line |
| 6       *fileLines* ← list from *filename* | 6       *fileIndex* ← index number from error line |
| 7       *fileIndex* ← index number from error line | 7       **if** *fileName* = ".f" or *fileName* = ".for" **then** |
| 8       **for** *ii* in range of *fileIndex* **do** | 8         *fileLines* ← list from *fileName* |
| 9         **if** "IMPLICIT NONE" in *fileLines[ii]* **then** | 9         **for** *ii* in range of fileLines **do** |
| 10           *implicitIndex ← ii* | 10           **if** *ii* ≠ *fileIndex* **then** |
| 11         **end if** | 11           write *fileLines[ii]* to file |
| 12       **end for** | 12         **else if** "REAL/INTEGER" in *fileLines[ii]* **then** |
| 13       **for** *iii* in range of *fileIndex* **do** | 13           **if** "," in *fileLines[ii]* **then** |
| 14         **if** *iii* = *implicitIndex* **then** | 14            write *fileLines[ii]* to file |
| 15           write "" to file | 15           **else** |
| 16         **else if** *iii* = *fileIndex* **then** | 16            write "C Deliberately left commented" to file |
| 17           write *fileLines[iii]* to file | 17           **end if** |
| 18           write "IMPLICIT NONE" to file | 18         **else** |
| 19         **else** | 19           write *fileLines[ii]* to file |
| 20           write *fileLines[iii]* to file | 20         **end if** |
| 21         **end if** | 21       **end for** |
| 22       **end for** | 22     **end if** |
| 23     **end if** | 23     **end if** |
| 24   **end for** | 24   **end for** |
| (C) Algorithm 5: Fixing Fortran error 6362-7319 | (D) Algorithm 6: Fixing Fortran error 6222 |
| 1   **for** *i* in range of Fortran file **do** | 1   **for** *i* in range of Fortran file **do** |
| 2     *A ← list[i]* | 2     *A ← list[i]* |
| 3     *index1 ← A*.find("error#6362") | 3     *Index ← A*.find("error#6222") |
| 4     *index2 ← A*.find("error#7319") | 4     **if** *index* ≠ −1 **then** |
| 5     **if** *index1* ≠ −1 or *index2* ≠ −1 **then** | 5       *filename* ← Fortran file name from error line |
| 6       *Var* ← variable name from error line | 6       *fileLines* ← list from *filename* |
| 7       *filename* ← Fortran file name from error line | 7       *fileIndex* ← index number from error line |
| 8       *fileLines* ← list from *fileName* | 8       **for** *ii* in range of *fileLines* **do** |
| 9       **for** *ii* in range of *fileLines* **do** | 9         **if** *ii* = *fileIndex* and "IMPLICIT NONE" in *fileLines[ii]* **then** |
| 10         **if** "REAL/INTEGER" + *var* in *fileLines[ii]* **then** | 10           write a new line to file |
| 11           write a new line to file | 11         **else** |
| 12         **else** | 12           write *fileLines[ii]* to file |
| 13           write *fileLines[ii]* to file | 13         **end if** |
| 14         **end if** | 14       **end for** |
| 15       **end for** | 15     **end if** |
| 16     **end if** | 16   **end for** |
| 17   **end for** | |

**Figure 13.** The pseudocodes for cleaning Fortran compiler errors: (**A**) 6278 (USE statement positioned incorrectly), (**B**) 6418 (name already assigned), (**C**) 6362 (invalid data types of arguments) and 7319 (procedure assumed external), and (**D**) 6222 (incorrectly positioned IMPLICIT statement).

## 5. Discussion

The initial development of the ForDADT project involved using ifort to compile a list of Fortran errors and a file solution that resolves the issues from the BuildLog. However, this setup does not consider cascading issues when compiler errors occur. In this case, the program runs into the issue of continuously reiterating its execution until all errors are fixed. Due to the manual process of compiling ifort repeatedly, this configuration can be exponentially expensive when dealing with copious number of files and thus creates a necessity to redesign the compiler.

Available tools such as Phortran [42] and fprettify [43] provide auto formatting for modern Fortran. What ForDADT promises beyond available tools is automating the error fixing by avoiding the compiler constraints for large-scale software. ForDADT has a more modular solution to construct the BuildLog, and, by using an experimental Fortran reader, it can reproduce compiler errors without the use of ifort. Pattern analysis method used in the findError function (Figure 14) is critical to ForDADT as it helps detect issues identical to the compiler.

| **Algorithm 7: Checking Fortran error 6404** |
|---|

| | |
|---|---|
| 1 | *varList* ← to store declared variables |
| 2 | *exList* ← to store library variables |
| 3 | **for** *i* in range of Fortran file **do** |
| 4 | A ← *list*[*i*] |
| 5 | /* Ignore any of the following comment lines */ |
| 6 | **if** "C" or "*" or "!" in *A* **then** |
| 7 | skip the comment line |
| 8 | **end if** |
| 9 | **if** string literals in *A* **then** |
| 10 | Remove the literals |
| 11 | **end if** |
| 12 | /* Function blocks that strip the excess data from the code */ |
| 13 | strip the declared variables not using FORTRAN 77 styling |
| 14 | strip FORMAT functions |
| 15 | strip function names |
| 16 | replace all the characters not in {a-z, A-Z, 0-9, \n, \-, \., \} with a comma |
| 17 | **end for** |
| 18 | *leftIndex* ← 0 |
| 19 | *rightIndex* ← 0 |
| 20 | **while** *leftIndex* ≠ −1 **do** |
| 21 | **if** *leftIndex* = 0 and *rightIndex* = 0 **then** |
| 22 | *rightIndex* ← find index of "," |
| 23 | **else if** *leftIndex* ≠ −1 and *rightIndex* = −1 **then** |
| 24 | /* if the comma does not exist, set index to the end of line */ |
| 25 | *rightIndex* ← end of line index |
| 26 | **end if** |
| 27 | /* Use comma as separator to split multiple variable declarations into separate ones */ |
| 28 | **if** *leftIndex* = 0 and *rightIndex* = −1 **then** |
| 29 | /* Check single line variable */ |
| 30 | **if** variable was not previously declared **then** |
| 31 | write "error#6404" to file |
| 32 | **end if** |
| 33 | /* Check the first variable */ |
| 34 | **else if** *leftIndex* = 0 **then** |
| 35 | **if** variable was not previously declared **then** |
| 36 | write "error#6404" to file |
| 37 | **end if** |
| 38 | /* Check the middle and last variables */ |
| 39 | **else** |
| 40 | **if** variable was not previously declared **then** |
| 41 | write "error#6404" to file |
| 42 | **end if** |
| 43 | increment *leftIndex* |
| 44 | increment *rightIndex* |
| 45 | **end while** |

**Figure 14.** The extended algorithm for finding Fortran compiler error 6404 (explicit type missing).

In the instance of findError6404, the program checks for declared variables, and pattern analysis is used excessively to sort the Fortran files into compatible data. By isolating keyword function statements such as IF/ELSE and FORMAT, it is possible to sort statements into individual variables and check if those are properly declared (Figure 14). ForDADT simplifies the compilation process for detecting error cases in Fortran, designed to identify general syntax errors.

Figure 15 shows the computation time of error fixing for six different codes based on the number of lines in the code.

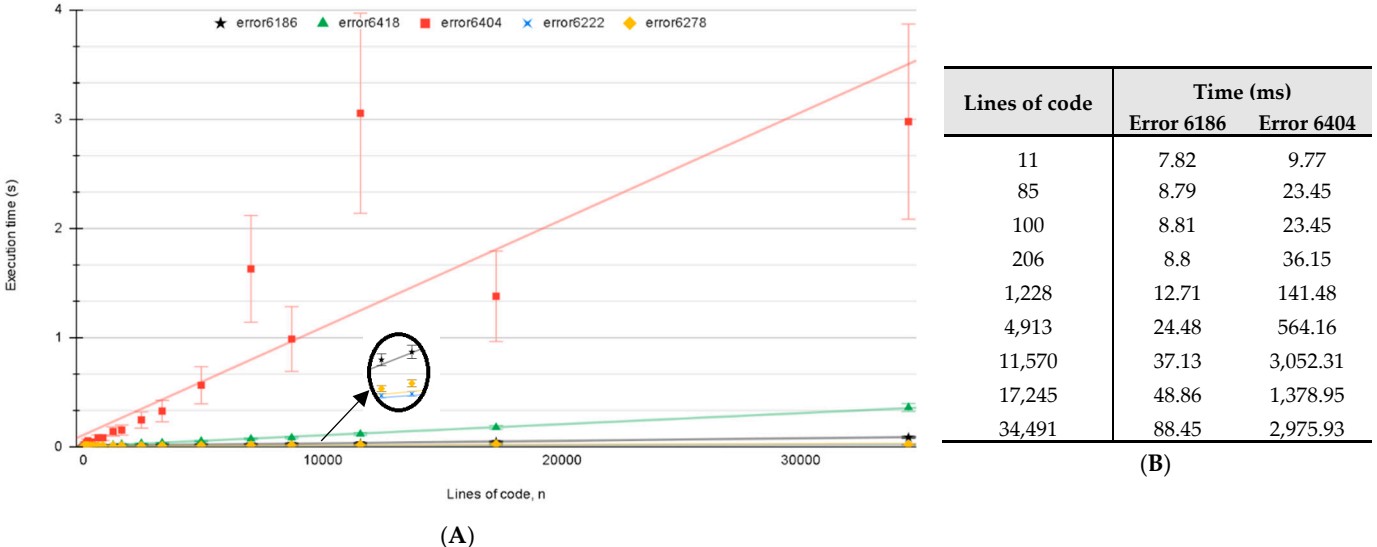

| Lines of code | Time (ms) | |
|---|---|---|
| | Error 6186 | Error 6404 |
| 11 | 7.82 | 9.77 |
| 85 | 8.79 | 23.45 |
| 100 | 8.81 | 23.45 |
| 206 | 8.8 | 36.15 |
| 1,228 | 12.71 | 141.48 |
| 4,913 | 24.48 | 564.16 |
| 11,570 | 37.13 | 3,052.31 |
| 17,245 | 48.86 | 1,378.95 |
| 34,491 | 88.45 | 2,975.93 |

(A)

(B)

**Figure 15.** (**A**) Computation time of the ForDADT error reader for different errors based on the number of processed lines. (**B**) Time comparison for catching errors 6186 and 6404.

The time complexity for the error fixing subroutines is O(n) as they mainly follow a linear search algorithm. Computations were performed on AMD Ryzen 5900X, 12-Core Processor @ 3.70 GHz, and 32 GB RAM. Figure 15A illustrates that ForDADT's error algorithms each compile in a linear time such that it is proportional to the lines of code in the file. There is small variation from the linear approximation for the error sets 6186, 6222, 6278, and 6418. Error 6404 takes in account external file declarations when dealing with source variables in the file. As each subroutine may need different sets of external file declaration, the computation time of the file in question increases, as seen in Figure 15B.

## 6. Conclusions

In this paper, we documented the development of a prototype tool to automate the manual labor of refactoring the individual files. ForDADT is a Python program used to process Fortran files and automate the cleaning for compilation errors. ForDADT project removes Fortran errors and updates the codes written in older versions to version 19.0.5.281. Automating common error cases from the Fortran program reduces the amount of refactoring necessary when compiling. This project has been developed to bridge the refactoring and automation processes to help improve developers' efficiency when refactoring code. The developed tool automatically updates thousands of Fortran files and builds the software to find and fix the errors using pattern matching and data masking algorithms. These upgrades produce a more readable Fortran program that includes safeguards to prevent accidental mistyping of variables and unintentional changes during program execution.

In our future work for this project, we will aim to create a dynamic analysis application to assist immediate error detection and notify developers when syntax errors can be corrected. This adaptive error modeling has potential applicability for a better suited robust design in error proofing the automation process.

**Author Contributions:** Conceptualization, L.M. and P.T.; methodology, L.M.; software, L.M. and P.T.; validation, L.M.; formal analysis, L.M.; investigation, L.M.; writing—original draft preparation, L.M.; writing—review and editing, P.T.; supervision, P.T.; project administration, P.T. All authors have read and agreed to the published version of the manuscript.

**Funding:** This research received no external funding.

**Data Availability Statement:** The tool developed in this project is open source. The codes are available at https://github.com/LMAK00/Fordadt.

**Acknowledgments:** The authors acknowledge that this project was carried out on the traditional lands of the Coast Salish peoples, including the unceded territories of the xʷməθkʷəy̓əm (Musqueam), Sḵwx̱wú7mesh (Squamish), and səlilwətaɬ (Tsleil-Waututh) Nations.

**Conflicts of Interest:** The authors declare no conflict of interest.

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
