# Peer review of "An Automated Tool for Upgrading Fortran Codes"

_2674-113X, doi:10.3390/software1030014_

Round 1
Reviewer 1 Report
The paper is well written and the topic is relevant and of interest for scientific community. The results could be useful for both: future research and current practice. However, there are some questions that are to be addressed before the paper can be published.
1) The chapter referring to the results of previous research should be added. The introduction chapter does provide a short review on important concepts, but recent and relevant papers related to the research context are missing.
2) The Methodology chapter describes the author’s approach to the solution of the problem in detail, but does not argue the choice of methodology. Are there any other strategies and approaches that could be used? How was the decision on selected methodology made?
3) Authors need to explain the contribution of their research compared to the results of previous research. How does the developed tool differ from existing similar tools? What are the benefits of using FortADT compared to similar tools?
4) Future work should be discussed.
Author Response
Dear respected reviewer,
We would like to thank you for your thorough and detailed review of our submitted draft. All your comments and feedback were to the point and improved the quality of our work significantly. We really appreciate your time and efforts.
In this revised submission, we did our best to apply most of your suggestions and clarify the confusions. We have tried to answer to your feedback in the attached file. We have also uploaded the revised version of the paper. All are changes are highlighted.
Best regards.

Reviewer 2 Report
# Software article review
The paper describes a prototype project for automating part of the refactoring/compilation/debugging pipeline for large-scale Fortran code bases. The authors do a reasonable job of defining the problem and outlining the requirements for an appropriate solution. The explanation of the operation of the tool is, however, stuck in an unhappy middle ground between too much and too little detail.
As a summary of the major problems fixed by the tool, the paper is too detailed. Paragraphs of text explaining how the order of variable declarations, USE, and IMPLICIT NONE statements is analysed and corrected (for instance) do not help the reader understand the benefit of the tool. If the point is to demonstrate how the automatic debugging of minor errors in syntax saves time or improves code quality, then evidence in the form of actual numbers or exemplars should be provided.
The sample program with pruned results (Fig 7), for instance, does not actually show me anything useful. I can see how the code has been prepared for the verification stage, but I would consider the actual result here to be the code in its final form, after any errors have been fixed. These kinds of exemplars would be one kind of result which I would consider valuable to the reader. The other would be headline figures (estimated or otherwise) for how much time is saved by using the tool. Even an example figure for how many errors are caught and corrected by the tool would be useful as then the reader could appreciate the scale of the manual task vs the automated.
On the other hand, as a document detailing the operation of the software, the paper is sorely lacking in detail. To be clear, I do not think a more detailed analysis of the various algorithms etc. would be useful; I can't see the use of such a document. But I am simply making the point that the text in its current form is not sufficiently clear to justify its brevity, nor sufficiently rigorous to justify its length.
I will admit to being quite excited by the abstract and introduction of the paper. This sounds like a tool that would be incredibly useful to me and my community in our own research. However the paper does not deliver on that promise for the reasons outlined above. More pressingly, however, there is absolutely no way for me to test the claims. The software has not been made available with the paper, nor did an extensive search online yield any results. Thus none of the claims in the paper is verifiable, and none of the 'results' is reproducible, although, as hinted above, I actually cannot find any results in the paper.
Hence I am unable to recommend the article for publication at this time. Given the potential importance of such a project, I would be very keen to see a revised manuscript. However, unless I am actually allowed to see and run the software for myself, I will be unable to recommend the piece for publication.
Some specific comments follow.
Lines 43-45
> A moderate size of the correspondents automates about 30% of the testing phase [7] and there is a ceiling for the degree of software automation in the testing environment.
It is not clear what is meant by "a moderate size of the correspondents". Is this a proportion of a group of people? If so, who are these correspondents, and with whom are they corresponding?
Lines 99-117
In the list of modifications for modern Fortran code, it would be useful for the reader to have the references attached to the specific modifications. I.E I would like to see
COMMON block elimination [ref]
EQUIVALENCE statement elimination [ref]
So that readers more easily find relevant literature/tools for specific problems in their own Fortran code.
Line 126-127 (and again on lines 377-378)
It is not clear to me what it means to update a code to a compiler version. Do you mean that by carrying out the modifications using your tool, the code is guaranteed to compile with that compiler?
Line 132
Is the dilemma with the specific (intel) compiler you just mentioned, or with all Fortran compilers?
Line 173
> ensuring that the previous errors are properly compiled
Do you mean that the bugs are fixed, or that the information on errors is compiled into some useable, reliable form?
Figures 4 & 5
Images of algorithms, pseudocode and code samples are hard to view. Can they be converted to some form of text?
Lines 262-269
From this it is not clear how variables from USEd libraries will be handled. The text states that it will search in the root directory for external files. But if I have something like USE MPI in my module/subroutine, then I may use variables from that library, and (from this description at least) these won't be caught by the scan for variable statements.
Figure 9
State what each of the compiler errors is so that it is easier to interpret what the algorithm is attempting to do.
Author Response

(The authors gave the same response as above.)

Round 2
Reviewer 2 Report
Thank you to the authors for making the requested changes, which without doubt have improved the manuscript and alleviated some of my concerns. However, there remains a fundamental and rather ironic problem with this submission which has been elucidated by the authors' provision of their software (which, inexplicably is housed at FordADT, rather than FortADT at GitHub, but never mind).
The problem is this: the authors express the ultimate aim of improving the quality of Fortran code. Such an effort is unquestionably important. However, the FortADT software itself is provided without documentation, has not even got a README file to explain how it works, and contains a high degree of what I would consider to be rather sloppy programming. For instance the very first section of the main.py file imports the path object from the os module in three different ways, there are multiple instances of objects imported and not used, and there isn’t a doc-string in sight. In the drive to improve code quality, it would be bizarre for the tool itself to be poorly written, documented and maintained.
Moreover, (and perhaps this is a symptom of the fact there were no instructions for use), the tool failed to catch basic errors when I tried to test it.
This is a discussion not well suited for Peer review of a paper. I would be delighted to engage with the authors by logging GitHub issues for these problems, but that will have to wait until after this review process is completed.
All of that said, if I am judging the quality of the paper, and with the assumption that the code can (and will) be cleaned up, improved and any bugs fixed, then I think there are only three minor modifications necessary to raise the paper to the appropriate standard for publication.
-
My earlier suggestion to include an explanation of each error code was specific to Figure 9 (now 12). While the addition of table 3 is useful in general, the (now) figure 12 caption should be expanded to read (something like)
Figure 12. The pseudocodes for cleaning Fortran compiler errors: (A) 6278 (USE statement positioned incorrectly), (B) 6362 (invalid data types of arguments) and 7319 (procedure assumed external), (C) 6222 (incorrectly positioned IMPLICIT statement), and (D) 6418 (name already assigned)
- The inclusion of Figure 15 is welcome: it certainly gives some sense of the scale of the computational effort. However, this is not what the tool is for. It is claimed that FortADT can automate what would otherwise be an arduous manual process. However, the computational time is not a measure of the effectiveness of that automation, as the effort required for extensive pattern matching massively increases the number of operations compared to what a human developer would be doing (e.g. spotting typos). There is no perfect metric to judge this, but I suggest a simple number: in the cases reported in Figure 15, how many errors are located and fixed by FortADT in 30000 lines of code? If I then see that this number of errors can be dealt with in a matter of seconds by FortADT, then I can use my imagination to work out how long that would take me to fix manually.
- The new paragraph describing future work is fine, but I’d like the authors to say more. As a Fortran developer, I need to know how this proposed dynamic analysis application fits into the broader context of development tools. In particular, we already have tools for locating syntax errors: compilers and text editors. What FortADT promises beyond those tools is what should be discussed here. (Automating the fixing of those errors seems to be the unique selling point, so emphasise that.) I would also like the authors to reflect on what tools are available for other languages, that are currently missing for Fortran. E.G the python project fprettify is an auto-formatting tool for modern Fortran. How would the authors envisage FortADT being used or expanded in concert with tools like fprettify and others to round out the toolset for Fortran developers?
I don’t wish to review this paper again, as, with the above modifications, I think it will serve its purpose. However, if the authors wish it to be effective, then the code it points to needs to be improved in the ways I describe. I look forward to seeing the tool developed and used in the future.
Author Response
Dear respected MDPI Software reviewer and editor,
We would like to reiterate our appreciation for your thorough and detailed review. Our revised paper has a higher quality because of your thoughtful feedback. In this revised version, we addressed your new comments as well as two comments from the first round which we promised to address later.
Please be aware that the order of Figures has changed in the new revision. Attached are the list of the comments and our responses (first two comments are from the previous round).
Many Thanks and best regards.
